# Naturally-Occurring Tyrosinase Inhibitors Classified by Enzyme Kinetics and Copper Chelation

**DOI:** 10.3390/ijms24098226

**Published:** 2023-05-05

**Authors:** Hee-Do Kim, Hyunju Choi, Fukushi Abekura, Jun-Young Park, Woong-Suk Yang, Seung-Hoon Yang, Cheorl-Ho Kim

**Affiliations:** 1Molecular and Cellular Glycobiology Unit, Department of Biological Sciences, SungKyunKwan University, Seoburo 2066, Jangan-Gu, Suwon 16419, Republic of Korea; hdk0330@naver.com (H.-D.K.); hjchoi@skku.edu (H.C.);; 2Environmental Diseases Research Center, Korea Research Institute of Bioscience and Biotechnology, 125 Gwahak-ro, Daejeon 34141, Republic of Korea; 3Zoonotic and Vector Borne Disease Research, Korea National Institute of Health, Cheongju 28159, Republic of Korea; 4National Institute of Nanomaterials Technology (NINT), POSTECH, 77, Cheongam-ro, Nam-gu, Pohang-si 37676, Republic of Korea; 5Department of Medical Biotechnology, Dongguk University, Seoul 04620, Republic of Korea

**Keywords:** tyrosinase inhibitors, catalysis site structure, kinetics, copper chelation

## Abstract

Currently, there are three major assaying methods used to validate in vitro whitening activity from natural products: methods using mushroom tyrosinase, human tyrosinase, and dopachrome tautomerase (or tyrosinase-related protein-2, TRP-2). Whitening agent development consists of two ways, melanin synthesis inhibition in melanocytes and downregulation of melanocyte stimulation. For melanin levels, the melanocyte cell line has been used to examine melanin synthesis with the expression levels of TRP-1 and TRP-2. The proliferation of epidermal surfaced cells and melanocytes is stimulated by cellular signaling receptors, factors, or mediators including endothelin-1, α-melanocyte-stimulating hormone, nitric oxide, histamine, paired box 3, microphthalmia-associated transcription factor, pyrimidine dimer, ceramide, stem cell factors, melanocortin-1 receptor, and cAMP. In addition, the promoter region of melanin synthetic genes including tyrosinase is upregulated by melanocyte-specific transcription factors. Thus, the inhibition of growth and melanin synthesis in gene expression levels represents a whitening research method that serves as an alternative to tyrosinase inhibition. Many researchers have recently presented the bioactivity-guided fractionation, discovery, purification, and identification of whitening agents. Melanogenesis inhibition can be obtained using three different methods: tyrosinase inhibition, copper chelation, and melanin-related protein downregulation. There are currently four different types of inhibitors characterized based on their enzyme inhibition mechanisms: competitive, uncompetitive, competitive/uncompetitive mixed-type, and noncompetitive inhibitors. Reversible inhibitor types act as suicide substrates, where traditional inhibitors are classified as inactivators and reversible inhibitors based on the molecule-recognizing properties of the enzyme. In a minor role, transcription factors can also be downregulated by inhibitors. Currently, the active site copper iron-binding inhibitors such as kojic acid and chalcone exhibit tyrosinase inhibitory activity. Because the tyrosinase catalysis site structure is important for the mechanism determination of tyrosinase inhibitors, understanding the enzyme recognition and inhibitory mechanism of inhibitors is essential for the new development of tyrosinase inhibitors. The present review intends to classify current natural products identified by means of enzyme kinetics and copper chelation to exhibit tyrosinase enzyme inhibition.

## 1. Importance of Tyrosinase and Melanin Synthesis

Melanin has been known to be a causative factor for age spots, melasma, freckles, and chloasma. It has consequently been regarded as a major target for anti-melanogenic agents for the prevention of hyperpigmentation diseases. Pigmentation refers to coloring from a pigment melanin. Cells in the skin synthesize melanin if they are normal or healthy. Melanocytes generate melanins as protective agents of human skin against ultraviolet (UV) irradiation via melanogenesis. Skin pigmentation disorders affect skin color. A number of skin disorders are characteristic of melanin synthesis, including acanthosis nigricans, lentigines, incontinentia pigmenti, melanoma, cervical poikiloderma, albinism, melasma, pityriasis alba, periorbital hyperpigmentation, progressive pigmentary purpura, and Parkinson’s disease-associated neurodegeneration [1]. Melanin’s protective effects on UV-irradiated reactive oxidative species generation and wrinkling damages are evolutionarily adapted to life [2], but abnormal and hypermelanin pigmentation is harmful. Melanins are structurally associated with three distinctive colorful compounds: eumelanin, pheomelanin, and neuromelanin [3]. 

The melanogenesis pathway begins with L-tyrosine or L-DOPA oxidation as the starting material for dopaquinone by tyrosinase, and the second step results in the formation of quinone to serve as a substrate for subsequent steps that produce melanin. Three enzymes are required to initiate melanogenesis [4]. Tyrosinase is a central glycoprotein enzyme in the membrane region of the distinctive endosomal compartment, referred to as the melanosome. Tyrosinase (EC 1.14.18.1) catalyzes its substrates in a rate-limiting action mechanism to its melanogenesis reaction in the first two enzyme steps. The enzyme catalyzes the addition of the hydroxyl group from its substrate L-tyrosine to the intermediate 3,4-dihydroxyphenylalanine (DOPA) and the DOPA oxidation to yield its end-product DOPA-quinone (Figure 1). Tyrosinases, along with tyrosinase-related protein (TRP), catechol oxidases, and hemocyanins, belong to the type-III copper protein family, and tyrosinases are found in fungi, plants, and animals. Its catalysis of the conversion of L-tyrosine to L-DOPA is solely dependent on copper ions. Type-III copper oxidases have paired copper-binding sites of two copper Cu(A) and Cu(B) ions. Each copper ion is coordinated for bonding with three His residues in the enzyme catalytic site of tyrosinase. The tyrosinase catalytic site bears three distinct states of oxidization (*oxy*), reduction (*met*), and deoxidization (*deoxy*); the *met* and *oxy* enzyme states are ready to act as enzymatic catalysts of the diphenol substrates. The *oxy* enzyme state, but not the *met* enzyme state, is also ready to serve in the enzymatic catalysis of the monophenol substrates. The *deoxy* enzyme state combines with oxygen molecules. Thus, its active site with bi-copper ions has six His coordination sites, and this state potentiates monophenol oxidization by monophenolase activity (or cresolase) and diphenol oxidization by diphenolase activity (or catecholase) to form *ortho*-quinones and self-polymerized melanin. In the oxidation-reduction states, Cu_2_-O_2_ represents monophenolase and diphenolase enzyme activities, Cu_2_-O represents diphenolase enzyme activity, and Cu_2_ does not represent any enzyme activity, depending on the oxygen coordination amount. 

The melanosomal membrane isoform-I of tyrosine hydroxylase (referred to hereafter as THI) converts the L-tyrosine substrate to the L-DOPA intermediate product, and this step stimulates tyrosinase enzyme activation. The cytosolic enzyme form phenylalanine hydroxylase (PAH) converts L-phenylalanine residues to L-tyrosine in a cofactor 6-tetrahydrobiopterin (6BH4)-dependent manner [4]. Two THI and PAH proteins are similar to tyrosinase with approximately 40% homologies with TRP-1 and TRP-2 protein sequences. TRP-1 activates and stabilizes tyrosinase enzyme and melanosomal endosome formation, and it also increases the eumelanin vs. pheomelanin ratio and substrate peroxidation levels [5]. Mutations in TRP-1 cause hypopigmentation in the skin or hair. TRP-2 function shows dopacrome tautomerase enzyme (DCT; EC 5.3.3.12) activity with a cofactor of zinc metal instead of copper [4]. In eumelanin biosynthesis in mammals, DCT converts L-dopachrome—through its catalysis reaction—to its intermediate product of 5,6-dihydroxyindole-2-carboxylic acid. The DCT enzyme also known as TRP-2 belongs to the metalloenzymes, a group that includes three metallic enzymes. Figure 1 shows a systemic illustration of tyrosinase enzyme action on its substrates as well as the biosynthesis of the two different melanin forms [6].

The development of a whitening agent has thus far proceeded according to two methods, inhibition of the melanocyte’s melanin synthesis and regulation of melanocyte stimulation. The research directions in this area have thus far included the development of the inhibitor of tyrosinase, a melanin-synthetic enzyme, which modulates Cu^2+^-based active sites, and studies targeting Fe^2+^, TRP-1, and TRP-2’s essential roles as Fe^2+^-regulating agents. Melanocyte-stimulating factors include endothelin-1, melanocyte stimulating hormone (α-MSH), nitrogen oxide (NO), histamine, paired box 3, microphthalmia-associated transcription factor, pyrimidine dimer (pTpT), ceramide, and granulocyte macrophage colony-stimulating factor (GM-CSF) [7,8]. Modulation of the promoter region of the tyrosinase gene has recently emerged as another factor. Growth of epidermal-surfaced cells is upregulated by stem cell factor, melanocortin-1 receptor (MC1-R), and cAMP. Melanin synthesis is increased by melanocyte-specific transcription factor (MITF) for the transcriptional regulation of melanin and tyrosinase synthesis [9,10]. TRP-1 and -2 and tyrosinase are all active in melanin synthesis [11]. Previous studies investigating melanin levels have used melanocyte cell lines to examine melanin synthesis; TRP-1 and -2 expression levels have also been examined. Thus, examining these inhibitors of growth and melanin synthesis in gene expression levels represents another alternative method for whitening research. Therefore, there is a field of recent research covering the inhibition of tyrosinase enzyme activity as well as melanin synthesis in its entirety. 

For cellular signaling in melanogenesis, the initial receptor MC1-R has predominantly been activated by its relevant and endogenous agonists. MC1-R is a G protein that is coupled to adenylyl cyclase and predominantly expressed in dermal melanocytes. MC1-R regulates melanocytic differentiation and consequently determines the direction of skin phototype via interactions with adrenocorticotropic hormone (ACTH) and α-MSH. ACTH and α-MSH are melanocortins produced by enzymatic proopiomelanocortin (POMC) cleavage. Carboxypeptidase-1 produces ACTH and β-lipotrophin via its POMC cleavage, while carboxypeptidase-2β generates ACTH and β-endorphin via its POMC cleavage. α-MSH and ACTH possess the melanotropic tetrapeptide residues of His-Phe-Arg-Trp that are commonly required for melanin pigmentation [2,3,5]. To compensate for their pituitary production in melanogenesis, melanocytes and keratinocytes produce them on the skin. ACTH-caused Addison’s Disease, ACTH-generating Nelson syndrome tumors, and ACTH-sustained administration have all been shown to cause skin hyperpigmentation. POMC peptide-induced MC1-R activation deposits eumelanin pigments rather than pheomelanin pigments. MC1-R activators induce adenylate cyclase activity and consequently increase cAMP levels. Then, cAMP activates protein kinase A (PKA) activity. PKA catalyzes phosphor-CREB formation, which is crucial for a cAMP response element to upregulate specific transcriptional factors such as MITF via the downstream signaling of mitogen-activated protein kinases (MAPKs) [9,10]. MAPKs-driven phospho-MITF transcriptionally upregulates the melanin-associated gene expression known for Rab27a and melanosomal matrix protein Pmel17, consequently leading to eumelanogenesis (Figure 2).

Human tyrosinase inhibitors are particularly useful in pharmaceutics and cosmetics. The discovery of whitening agents began in the 1960s with the discovery of the monobenzylether of hydroquinone. Historically, the representative whitening agent has been L-ascorbic acid (vitamin C) due to its melanin synthesis inhibition. L-ascorbic acid inhibits melanin synthesis via the reduction of dopaquinone to L-DOPA by L-ascorbic acid. However, L-ascorbic acid is unstable in formulations used in cosmetics. To solve this problem, several L-ascorbic acid derivatives have been developed, including magnesium ascorbyl phosphate, ascorbyl palmitate, and ascorbyl stearate, but these agents are precipitated and oxidized during solubilization. Tyrosinase inhibitors have been purified from plants, and previous studies have mainly examined kinetic tyrosinase inhibition using mushroom tyrosinases. 

The currently known skin-whitening and anti-melanin molecules include arbutin, deoxyarbutin, hydroquinone, deoxyarbutin derivatives, resorcinol, vanillin, niacinamide, kojic acid, arbutin-mimic isotachioside, hydroquinone derivatives (α and β-arbutin), azelaic acid, L-ascorbic acid, ellagic acid, and tranexamic acid [12,13,14,15,16]. Among them, kojic acid, ascorbic acid, rucinol, and arbutin (which has been identified from microbium by Japanese Shiseido Cosmetics Co., Tokyo, Japan) are the representative inhibitors of tyrosinase. The most general tyrosinase inhibitors are arbutin and kojic acid in cosmetics, but they are also cytotoxic to normal cells and impenetrable into the dermal skin tissue. The usage of other oxidized leads of hydroperoxide and hydroquinone is prohibited due to their toxicity. Therefore, there is a need for agents that have a stable structure with dermal skin-permeable, easy-to-synthesize, non-toxic, and melanin-reducing properties.

**Figure 2 ijms-24-08226-f002:**
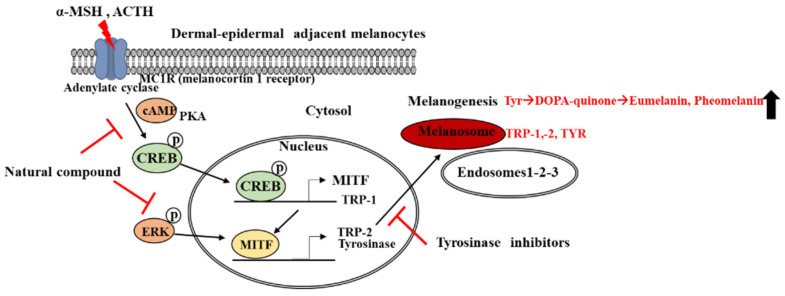
**Strategies of transcriptional inhibition in melanogenesis.** Modified from our previous publication (Park et al., 2021. [14]). 
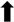
 indicates the increase of synthesis.

## 2. Inhibition of Melanin Biosynthesis and Natural Tyrosinase Inhibitor Sources

The inhibition of melanin biosynthesis reflects therapeutic and preventive options for melanogenesis. Tyrosinase inhibitors from natural products directly bind to tyrosinase and monophenolic tyrosinase proteins (Figure 3). A number of strategic approaches for reducing the melanin levels in dermal-epidermal junctional skin have been described, such as methods using direct enzyme inhibitors and gene downregulators. Because tyrosinase is a dinuclear or multi-copper-bearing protein, copper-chelation chemically blocks tyrosinase activities, and the enzyme also acts in a rate-limiting manner during catalytic melanin synthesis [17]. Tyrosinase uses L-tyrosine and diphenolic L-DOPA as substrates. *Deoxy*-tyrosinase easily binds to O_2_ to yield *oxy*-tyrosinase to serve as a monophenolase or diphenolase enzyme. The *met*-tyrosinase catalyzes as diphenolase, which oxidizes diphenols such as L-DOPA to quinone, and monophenolase, which oxidizes monophenols such as L-tyrosine to quinones such as L-dopaquinone, although tyrosinase activity is generally measured using its substrate, L-DOPA.

With regards to copper chelation, several natural products—including aromatic, phenolic, and poly-phenolic acids—are copper chelators. Reversely, these compounds can mimic the substrate of tyrosinase and consequently competitively inhibit tyrosinase. For example, tyrosinase competitively inhibits tyrosinase activity and downregulates melanin synthesis, and L-tyrosine also functions as a tyrosinase inhibitor [18]. Plants, bacteria, and fungi produce natural resources of antityrosinase agents with low cytotoxicity and high bioavailability, which are properties that are required for foods, cosmetics, and medicinal applications. Therefore, many tyrosinase inhibitors have been found in natural resources with a monophenole tyrosinase substrate and diphenol L-DOPA as a dopachrome substrate. 

### 2.1. Molecular Docking Simulation of Compounds with Tyrosinase and Structure–Activity Relationship (SAR) of Inhibitors

*In silico* computer-based molecular docking simulation of the quantitative relationship between structure and activity relationship (SAR) has served as an alternative approach to the bioactivity-guided fractionation, discovery, and identification of whitening agents in molecular models of inhibitors. Computational approaches using SAR-based theoretical simulation have been beneficial for inhibitor screening in many bio-informatic laboratories, although the technology is not comparable to the modern artificial intelligence chatbot. Computer simulation modeling predicts molecular candidates for functional skin susceptibility [19]. Quantitative (Q) SAR is suitable for molecular modeling with digital computational tools such as the Derek Nexus software [20]. 

In previous studies, the tyrosinase results have been experimentally obtained with TRP-1 (PDB ID: 5M8S) in the SWISS-MODEL (https://swissmodel.expasy.org/, accessed on 31 December 2022), and the 2D structures of candidate molecules have been formed with 3D structure conversions. The interaction energy between tyrosinase and molecules is then minimized using the ChemOffice program (http://www.cambridgesoft.com, accessed on 31 December 2022). The molecular docking simulations of the binding molecules to the tyrosinase enzyme are carried out using the Autodock Vina program [21]. The tyrosinase-binding pocket is decided by using predicted active sites obtained from the standard control fit for the tyrosinase-kojic acid binding (PDB ID: 5M8M) complex and the molecular dockings appear in Chimera form [22]. Based on the docking simulation outcomes, possible hydrogen bonding can be checked by using bonding relaxation constraints with 0.4 angstroms and 20.0 degrees as defaults in Chimera [23]. For example, a molecular docking simulation using human tyrosinase and Avn A to Avn C isolated from oats has shown that AvnC is a potent and selective tyrosinase inhibitor. AvnA-C is easily located at the binding sites of kojic acid complexed with tyrosinase (Figure 4). The phenylpropanoids are located inside the pocket. With a value of −4.0 kcal/mol, Avn C shows a stronger tyrosinase-binding affinity than AvnA (−3.7 kcal/mol) or AvnB (−2.9 kcal/mol), as calculated by AutoDock Vina [17].

### 2.2. Derek Nexus for Prediction of Skin Sensitization

Since 2013, animal testing has been prohibited in the pre-examination and product development stages of cosmetics. Since then, alternative models that do not involve animals have been considered and designed for such in silico simulations in nonanimal alternative and artificial in vitro and in silico models [19,20]. Currently, several software tools, such as TOPKAT, Derek Nexus, and Toxtree, are embedded with a classification database. The three in silico models are available using an algorithm modeling method. TOPKAT is based on statistics linked with QSAR models, Derek Nexus is a model based on knowledge, and Toxtree is based on the Bundesinstitut für Risikobewertung or Federal Institute for Risk Assessment. For regression- and classification-based models, QSAR analysis and skin-sensitizing predictions have not been able to achieve skin sensitization. Then, using QSAR-based approaches, the two artificial models have been combined with classifications and used for compound prediction by systemic categorization; these approaches are based on skin-responsive potentials to sensitization and regression for the predicted potency quantitation by Derek Nexus [20]. It is easy to predict which natural products will be non-sensitizers in terms of skin sensitization in mammals. When QSAR analysis is conducted, structural alerts are guided in terms of skin sensitization, unclassification, or misclassification. Using the QSAR system, the skin sensitization potential sub-structure can be calculated for its substituted analogs to decide its non-sensitizer property for human skin. In the SAR, the anti-tyrosinase activity comes from the heptadecenyl chain responsible for hydroquinone ring oxidation. 

## 3. Tyrosinase Inhibitor Types Based on Action Mechanism

### 3.1. Classical Tyrosinase Inhibitor Types 

Conceptional tyrosinase inhibitors are classified as inactivators and reversible inhibitors in anti-melanoma agents and cosmetics on the sole basis of the molecular recognizing properties of the enzyme. Tyrosinase inactivators are termed suicide inactivators as the substrates cause 3rd structural and conformational alterations of tyrosinase confirmation shifts and lead to enzyme inactivation [24]. 

### 3.2. Tyrosinase Inhibitor Types Based on Action Mechanism

Tyrosinase enzyme inhibitors are generally composed of four types. Based on their inhibitory action mechanisms, they are classified as competitive inhibitors, uncompetitive inhibitors, competitive/uncompetitive mixed-type inhibitors, and noncompetitive inhibitors. 

#### 3.2.1. Competitive Inhibitory Type 

Competitive inhibitors recognize and occupy the enzyme active site of a free enzyme in a solution to prevent the binding of its substrates to the enzyme active sites. For example, simple phenolic inhibitors include hydroquinones [1], deoxyarbutins [25], 4-6-hydroxy-2-naphthyl-1,3-bezendiol, and resorcinol (resorcin) [13]. Alkylhydroquinone 10′(Z)-heptadecenylhydroquinone is a tyrosinase inhibitor that is stronger than any known hydroquinone. As a tyrosinase inhibitor and substrate, hydroquinones such as α-/β-arbutins are a typical competitive type [26]. Phenolic compounds are other common inhibitors. Phenolic compounds that show tyrosinase inhibitory activities include simple phenols, polyphenols, flavonoids (flavones, isoflavones, flavones, flavanones, flavanoles, dihydroflavones, and anthocyanidins), and tannins. They possess one or multiple aromatic rings with a 9C-OH group or multiple -OH groups on their backbone structures. They are conjugated to saccharides or organic acids. For example, methoxy-hydroquinone-1-*O*-β-D-glucopyranoside and glycoside sotachioside are arbutin analogs. However, arbutin and isotachioside do not potently inhibit tyrosinase activity. By contrast, other glycoside sugars such as glucose derivative, xylose derivative, cellobiose derivative, and maltose derivative—all of which lack a methyl or benzoyl group in their central structures—potently inhibit tyrosinase activity. Among them, glucosides are prominently strong due to the glucosel-combined resorcinol [27]. 

Regarding flavonoids, major flavonoids can be divided into flavone, isoflavone, flavanol, flavonol, flavanone, and anthocyanidin, while known minor flavonoid forms include aurones, chalcones, coumarins, flavan-3,4-diols, dihydroflavones, and dihydrochalcones [28]. Flavonoid derivatives include prenylated or vinylated forms of flavonoid glycosides that are also known as tyrosinase inhibitors, such as kuwanon C, papyriflavonol A, sanggenon D, and sophoflavescenol, but isoprenyl group prenylation or flavonoid vinylation does not affect the tyrosinase inhibitory activity [29]. 3-*O*-β-galactosyl quercetin, 3′,5′-Di-*C*-β glucosyl phloretin, and galactosyl-3-myricetin are also known as flavonoid glycosides that inhibit tyrosinase activity [30]. Potenserin C and 3-*O*-α-L-rhamnosyl quercetin-2-gallate are competitive inhibitors of tyrosinase.

Flavones such as apigenin, chrysin, and luteolin do not chelate copper metals embedded on the tyrosinase enzyme, although the flavonols are chelated by the free 3-OH group [31]. Commonly known flavones include apigenin, baicalein, chrysin, luteolin, and flavone glycosides. Flavone derivatives include glycosidic forms of apigetrin, vitexin, and baicalin as well as polymethoxylated flavones of nobiletin and tangeretin [32]. Diphenolase inhibitors of tyrosinase include hydroxyflavones of baicalein, 6-hydroxyapigenin, 6-hydroxy-kaempferol, 6-hydroxygalangin, and tricin, which is also known as 5,7,4-trihydroxy-3,5-dimethoxyflavone [33]. Tyrosinase inhibitors include flavone, morusone, biflavone 3-*O*-β-D-glucosyl 4,5,5,7,7-pentahydroxy 3,3-dimethoxy 3,4-*O*-biflavone, flavone glucoside vitexin, and a *C*-glycosyl flavone isovitexin [34,35,36]. Flavones of kuwanon C, mormin, cyclomorusin, morusin, and norartocarpetin competitively inhibit tyrosinase [37]. Interestingly, because quercetin is not a cofactor, it is not a monophenolase inhibitor, while galangin is a monophenolase inhibitor, although it is a non-cofactor. Kaempferol as a non-cofactor is not a monophenolase inhibitor. By contrast, the three flavonols of quercetin, galangin, and kaempferol are diphenolase inhibitors through their copper chelation activities. Notably, 8-prenylkaempferol is a competitive inhibitor of tyrosinase [38]. Most flavonols belong to the competitive inhibitor type because they possess the 3-hydroxy-4-keto group in the flavonol backbone, which easily exhibits copper chelation of the enzyme active sites [31]. 4-*O*-β-D-glucosyl quercetin shows a strong tyrosinase inhibitory activity compared to the known inhibitory control of kojic acid [39]. The functionalization of the hydroxyl groups of the flavone skeleton coincides with the SARs of the inhibitory activity. The functionalization of aldehyde and methoxy as well as methoxyacetyl groups in benzaldehyde derivatives is also evidenced. However, flavonoid prenylation or vinylation modifications do not effectively confer tyrosinase inhibition, while flavonoid glycoside deglycosylation can enhance tyrosinase inhibitory activity. 

Isoflavones that are tyrosinase inhibitors include daidzein, daidzin, formononetin, genistein, glycitein, glabridin, glabrene, desmodianone H, glyasperin C, formononetin, odoratin, mirkoin, lupinalbin, mirkoin, genistin, texasin, tectorigenin, uncinanone B, and their glycosides of 2′-hydroxygenistein-7-*O*-gentibioside. *o*-Dihydroxyisoflavones with hydroxyl substituents at the aromatic ring along with 7,8,4′-trihydroxy- and 7,3′,4′-trihydroxy-isoflavones are tyrosinase inhibitors, but daidzein, glycitein, genistein, and 6,7,4′-trihydroxy-isoflavone exhibit low tyrosinase inhibitory activities [40]. 6,7,4′-Trihydroxyisoflavone is a competitive tyrosinase inhibitor of monophenolase that exhibits higher affinities than kojic acid [41]. However, their analog derivatives, such as daidzein, genistein, and glycitein, show little tyrosinase-inhibitory activity. Thus, the C-6 and C-7 hydroxyl groups of the isoflavone backbone have been suggested to be essential for tyrosinase inhibition. For example, isoflavone metabolic derivatives of 7,8,4′-trihydroxy- and 5,7,8,4′-tetrahydroxy isoflavones are irreversible inhibitors for both monophenolase and diphenolase enzymes [42]. 7,8,4′-trihydroxy- and 5,7,8,4′-tetrahydroxy-isoflavones are often referred to as suicide substrates of enzyme tyrosinase inhibition. The isoflavone OH groups located at the positions of C7 and C8 are essential for reversible and competitive inhibition as the irreversible suicide type. Isoflavone glabrene is an inhibitor of both monophenolase and diphenolase to tyrosinase [43]. Two isoflavones, uncinanone B and desmodianone H, inhibit tyrosinase activity, with uncinanone B exhibiting stronger inhibition than desmodianone H [44]. The isoflavones daidzein, glyasperin C, formononetin, genistein, mirkoin, texasin, tectorigenin, and odoratin inhibit tyrosinase activity [45]. Among the flavonoids, mirkoin reversibly and competitively inhibits tyrosinase more than the control, kojic acid [46]. Two isoflavonoids, lupinalbin and 7-*O*-gentibiosyl 2′-hydroxygenistein, are competitive inhibitors [47].

Among flavanones of eriodictyol, naringenin and hesperetin along with their glycosides such as hesperidin, hesperetin, liquiritin naringin, and flavanonol taxifolin as copper chelators are all reverse and competitive tyrosinase inhibitors. Hesperetin disrupts the tyrosinase structure via hydrophobic interactions and coordinates copper ions with three His residues located at the positions of HIS61, HIS85, and HIS259 in the enzyme active site pocket [48]. 6-isoprenoid-substituted flavanone [49] and steppogenin have both been shown to inhibit the enzyme [34]. Sanggenon-type isoprenyl flavanone, nigrasin K, and its derivatives sanggenon M/C/O, chalcomoracin, kuwanon J, and sorocein H are all tyrosinase inhibitors. Sanggenon D is much stronger than arbutin or kojic acid, which is often used as a positive control. Regarding flavan-3,4-diols and flavanoles, there are several known catechin derivatives of epicatechin, such as epigallocatechin, epicatechin gallate, epigallocatechin gallate, catechin, and proanthocyanidin [50,51]. The most complex flavonoids, which are also called tannins and known as flavan-3-ols, range from the catechin monomers and epicatechin isomers to the oligomeric and polymeric proanthocyanidin forms. Tannins and gallocatechin are tyrosinase inhibitors [36]. Catechin shows a higher tyrosinase inhibitory activity than a control, arbutin. Procyanidin-type proanthocyanidins competitively inhibit the tyrosinase monophenolase and diphenolase [52]. Moreover, (+)-catechin-aldehyde polycondensates chelate the tyrosinase active site to inhibit L-DOPA oxidation and L-tyrosinase hydroxylation [53]. (-)-8-Chlorocatechin has also been shown to be a competitive inhibitor of tyrosinase. 

The flavonoles that are known to inhibit tyrosinase activity are kaempferol, isorhamnetin, galangin, myricetin, quercetin, and morin, and their glycosides such as astragalin, rutin, and quercitrin. Kaempferol, quercetin, 4′-*O*-β-D-glucosyl quercetin, β-D-glucosyl 3-*O*-6-*O*-malonyl quercetin, β-D-glucosyl 3-*O*-6-*O*-malonyl-kaempferol, galangin, morin, (±)2,3-*cis*-dihydromorin, and 2,3-*trans*-dihydromorin all inhibit tyrosinase activity [31,54,55]. Morin inhibits tyrosinase activity in a reversible and competitive manner through multiple-phase kinetics and binding to a tyrosinase single site via interactions between H-bonds and van der Waals forces. Morin’s interaction with the tyrosinase enzyme induces 3D conformational shifts [55]. Galangin, kaempferol, and quercetin are specific tyrosinase inhibitors for L-DOPA oxidation via copper chelation [29,54]. Moreover, anthocyanidins including cyanidin, delphinidin, malvidin, pelargonidin, peonidin, and their glycosidic derivatives all inhibit tyrosinase activity [56]. For curcuminoids, the phenolic compounds of curcumin and desmethoxycurcumin are better tyrosinase inhibitors than kojic acid as the control. Curcumin competitively inhibits the enzyme. The curcumin 4-hydroxyl groups at the positional locations of C-2 and C-4- or C-3 and C-4-dihydroxyl phenolics inhibit the enzyme activity more strongly than kojic acid [57].

The derivatives of coumarins that are known to inhibit enzyme activity are 3-aryl form, 3-heteroaryl form, esculetin, cardol-form, phenyl form, hydroxyl form, 8-*epi*-cleomiscosin, umbelliferone, thiophosphonic acid diamides, diazaphosphinanes form, and resveratrol-hybrid form [58,59,60,61]. Among the hydroxyl forms of coumarin, 3-hydroxyl-coumarin and 7-hydroxyl coumarin highly inhibit enzyme activity whereas 4-hydroxyl coumarin has no inhibitory activity [61]. *Trans*- and *cis*-*N*-coumaroyl tyrosinaseamine potently inhibit tyrosinase activity. Butein, 1,3-diphenyl-2-propen-1-one, carthamin, phloretin, and sappan-chalcone are chalcones and dihydrochalcones that are flavonoids with tyrosinase inhibitors. Naturally occurring chalcones that inhibit tyrosinase activity are isoliquiritigenin with 2,4,4-trihydroxychalcone and glabrene, 2,4,2,4-hydroxycalcone and its 3′-substituted resorcinol derivatives, 2,4,2′,4′-tetrahydroxychalcone 2,4,2′,4′-tetrahydroxy-3-3-methyl-2-butenyl-chalcone, vulpinoideol-B, dihydrochalcone, morachalcone-A, and bavachinin [34,62]. In the SAR, OH- groups on both aromatic backbones located on chalcone and the hydroxyl number are crucial for tyrosinase inhibition, while catechol moiety does not influence tyrosinase inhibition [63]. The hydroxyl group locations on both aromatic backbones and hydroxyl number are also essential for the inhibitory efficacy in cholcone derivatives. Aurones such as Z-benzylidenebenzofuran-3,2H-one inhibit tyrosinase activity, although the tyrosinase inhibition of aurone itself is weak, while its derivatives with two or more OH- groups at the locations of C4, C6, and C4′ exhibit enhanced tyrosinase inhibition. In fact, the active aurone in the structure of 4,6,4′-trihydroxyaurone shows higher inhibition than kojic acid [64]. Among aurones, naturally-occuring 2-arylbenzofuran, 2R-2,3-dihydro-2-1-hydroxy-1-methylethyl-2,6-bibenzofuran-6,4-diol, benzofuran flavonoid mulberrofuran G, albanol B, and macrourins E are potent tyrosinase inhibitory natural products [65,66].

Phenolic acids can be grouped into two representative structures of hydroxybenzoates and hydoxycinnamates. Hydroxycinnamates include caffeic acid (CA), ferulic acid, and *p*-coumaric acid as the most common molecules. Benzoic acid propyl gallate, chlorogenic acid as CA ester, *p*-hydroxybenzoic acid, protocatechuic acid as dihydroxybenzoic acid, protocatechualdehyde, orsellinic acid as 2,4-dihydroxy-6-methylbenzoic acid, vanilic acid as 4-hydroxy-3-methoxybenzoic acid, *p*-coumaric acid, *m*-coumaric acid and its derivatives, CA and its *n*-nonyl ester, ferulic acid, 4-hydroxy cinnamic acid, and hydroxycinnamoyl derivatives all show tyrosinase inhibitory activities. Among the above inhibitors, propyl gallate is a reversible and mixed type diphenolase inhibitor. Meanwhile, uncompetitive type inhibitors are *n*-butyl, *iso*-propyl orsellinate, *n*-hexyl orsellinate, *n*-pentyl orsellinate, *n*-octyl orsellinate, and *sec*-butyl orsellinate, while tyrosinase activators are ethyl, methyl, *n*-cetyl, *tert*-butyl, and *n*-propyl orsellinates. Therefore, the inhibition trend of tyrosinase increases depending on chain length elongation, which suggests that the active site of the enzyme easily accepts 8-C alkyl chains [66,67,68,69,70,71,72]. For stilbenes, the known stilbene resveratrol derivatives include resveratrol and its derivatives and oxyresveratrol, azo-resveratrol, azo-oxyresveratrol, *E*-2-2,4-dihydroxyphenyl, diazinyl, phenyl-4-methylbenzenesulfonate, *trans*-resveratrol, and resveratrol dimer gnetin-C derivatives [56,73,74,75,76]. Resveratrol and oxyresveratrol exhibit superior inhibition of tyrosinase activity, but oxyresveratrol is cytotoxic to melanocytes. Moreover, several hydroxystilbene compounds show tyrosinase inhibition activity [77]. Although it inhibits L-tyrosinase oxidation, resveratrol is not a diphenolase inhibitor of tyrosinase. Resveratrol is both a tyrosinase monophenolase and diphenolase inhibitor, but the addition of L-cysteine decreases the inhibition level, thus pointing to the suicide inhibition mechanism of resveratrol [78]. 

Turning to lignans, they are complex in their chemical structures, which are generated from three original precursors. Lignans and lignan glycosidic derivatives are potent tyrosinase inhibitors. The tyrosinane inhibitors among terpenoids are monoterpenoid phenol, carvacrol, and its derivatives; bakuchiol; and iridoid glucosides and bis-iridoids of sylvestrosyl 7-*O*-caffeoyl-I and sylvestrosyl 7-*O*-*p*-coumaroyl-I [79]. Among the terpenoids, bakuchiol is a potent inhibitor. Meanwhile, quinones are derived from aromatic benzene or naphthalene. Glycosidic anthraquinone-C, alloin, senna, rhubarb, anthraquinones, and tanshinone IIA all inhibit tyrosinase activity [80,81,82]. Kojic acid, which is a well-known tyrosinase inhibitor, effectively inhibits pigmentation and oxygenation when tyrosinase oxidizes dopamine, DL-DOPA, and norepinephrine [83]. From alga including *Scotinophara latiuscula*, bromophenolic compounds of 2,3,6-tribromo-4,5-dihydroxybenzyl methyl alcohol, its methyl ether, and *bis*-2,3,6-tribromo-4,5-dihydroxybenzyl methyl ether have been isolated as tyrosinase inhibitors. Bromophenols have been shown to be a competitive-type inhibitor for L-tyrosinase with moderate activity. Molecular docking simulation has shown that the inhibitors bind to catalytic hydrogen and interact with halogen ions for the inhibitors [84]. Aromatic cinnamon and bay have also been shown to be reversible and competitive-type inhibitors [85]. T1 compound and bis-4-hydroxybenzyl sulfide is a competitive-type inhibitor to tyrosinase that is comparable to kojic acid. Molecular modeling analysis has shown that the sulfur atom’s T1 coordination with copper ions at the enzyme active site represents a key inhibitory activity [86]. The competitive-type tyrosinase inhibitors have been summarized in Table 1.

#### 3.2.2. Uncompetitive Inhibitory Type

Uncompetitive inhibitors only recognize the enzyme-substrate complex (ESC). For example, deoxyarbutin is a reversible tyrosinase inhibitor [1]. Luteolin, as a main component of Ginseng and *Moringa oleifera* extracts, has shown an uncompetitive inhibitory mode in diphenolase activity inhibition. Luteolin has been shown to be a main component and a distinct tyrosinase diphenolase ES complex inhibitor has been shown to be an uncompetitive type inhibitor by the kinetics of luteolin [87]. Valonea tannin isolated from acorn shells shows tyrosinase inhibition, and the kinetic inhibition mode of tyrosinase is a mixed competitive and uncompetitive type inhibitor. Based on Stern-Volmer fluorescence quenching, isothermal titration, and molecular docking simulation, valonea tannin is known to non-selectively bind to the tyrosinase surface via hydrophobic interactions and hydrogen bonds. Valonea tannin chelates a copper ion for inhibition activity in the SAR of valonea tannin, coupled plasma-optical emission, and radical scavenging [88]. *Asphodelus microcarpus* extracts also directly inhibit tyrosinase activity, and ethanolic extracts kinetically act as uncompetitive inhibitors. As constituents, chlorogenic acid, luteolin derivatives, naringenin, and apigenin reduce cellular tyrosinase activity and melanin synthesis in a manner comparable to kojic acid [89].

Naturally occurring acetophenone molecules of 2,5-dihydroxyacetophenone (DHAP) and 2,6-DHAP are murine tyrosinase inhibitors of the uncompetitive type. 2,5-DHAP is a strong inhibitor of both melanin biosynthesis and tyrosinase activity [90]. β-D-Glucosyl 3,4-dihydroxy-5-methoxybiphenyl-2-*O* and 3,6-dihydroxy-2,4-dimethoxy-dibenzofuran are inhibitors of tyrosinase activity, tyrosinase and TRP-2 expression, and paired box 3 and MITF mRNA expression. An enzyme kinetic assay showed that β-D-glucosyl 3,4-dihydroxy-5-methoxybiphenyl-2-*O* is an uncompetitive mixed-type inhibitor when l-3,4-dihydroxyphenylalanine is used as the substrate [91]. A phenolic glycoside compound, β-D-glucosyl Ov-16-4-3,4-dihydroxybenzoyloxymethyl phenyl-O, shows hypopigmentary activity on mushroom tyrosinase. Its tyrosinase inhibitory kinetics using L-DOPA oxidation indicate that it is an uncompetitive-type inhibitor [92]. Cinnamic acid, aloin, and sophorcarpidine all inhibit tyrosinase activity, and sophorcarpidine is an uncompetitive-type inhibitor, while cinnamic acid or aloin is a mixed-type inhibitor [93]. The uncompetitive type of tyrosinase inhibitors have been described in Table 2.

#### 3.2.3. Mixed Type Inhibitors with Competitive and Uncompetitive Modes

The mixed inhibitory type binds to both free enzyme and ESC forms. For example, cinnamic acid hydroxypyridinone derivatives and phthalic acid derivatives belong to the mixed inhibitor type for monophenolase inhibition [94]. Similarly, D-arabinose [95], brazilein [96] and thymol derivatives [97] are mixed-type inhibitors of di-phenolase. Baicalein is a mix-type inhibitor via a single binding site and spontaneous binding via van der Waals forces and H-bond binding. Circular dichroism spectral analysis indicates that the bonds decrease t α-helix level. According to a molecular docking simulation of tyrosinase, baicalein binds to the Met280 residue in tyrosinase [98]. Proanthocyanidins is a mixed competitive tyrosinase inhibitor with anti-melanogenic activity. The OH group on the proanthocyanidin B ring chelates the enzyme di-copper ions. Condensed tannins such as procyanidin, prodelphinidin, and propelargonidin, as well as acyl derivatives such as galloyl benzoate and *p*-hydroxybenzoate, are all reversible and mixed-type competitive inhibitors [99]. 3-Phenylbenzoic acid as a pheloic acid type is the most potent noncompetitive inhibitor to monophenolase and the most potent mixed-type inhibitor to tyrosinase diphenolase activity. However, its esterification abrogates the inhibitory activity of tyrosinase [100].

2-*S*-Lipoyl-CA, which is also known as CA-dihydrolipoic acid *S*-conjugate, is a mixed-type inhibitor of tyrosinase [101]. Flavonolignans consist of isosilybin A/B, silydianin, 2,3-dihydrosilychristin, silybin, and silychristin-A/-B. The isolated flavonolignans are inhibitors of both tyrosinase monophenolase and diphenolase. In kinetic studies, flavonolignans have shown the mixed-type I (K_I_ < K_IS_) inhibitory modes [102]. Natural tyrosinase inhibitors such as glycolic acid are mixed-type inhibitors that operate via glycolic acid binding to tyrosinase by hydrophobic attraction, thus causing a conformational shift in the enzyme molecules. Tyrosinase activity is inhibited by its active site (His 263) perturbation [103]. *E*-2-isopropyl-5-methylphenyl-3-4-hydroxyphenyl acrylate and its methoxyphenyl acrylate are strong mixed-type inhibitors [104]. Tannin-condensed forms include procyanidins, prodelphinidins, and their gallates, which are all reversible- and mixed-type inhibitors. Among the components, procyanidins are dominant molecules that serve as strong inhibitors to the monophenolase and diphenolase enzymes [105]. Essential oils derived from aromatic plants of *Magnolia officinalis* are reversible- and mixed-type inhibitors. Sesquiterpenoid agarozizanol E is a mixed-type inhibitor for tyrosinase [106]. Kinetic studies of hop tannins have shown that the tannins were competitive-uncompetitive mixed-type inhibitors. In silico molecular docking analysis has shown that the tannins bind to the tyrosinase active site via electrovalent and hydrogen bonding. Fluorescence quenching and free radical scavenging analysis have also proven that the tannins chelate copper ions [107]. 2-2-Phenylethyl chromones and sesquiterpenoids act in similar tyrosinase inhibitory modes. The mixed competitive and uncompetitive types of tyrosinase are summarized in Table 3.

#### 3.2.4. Noncompetitive inhibitory type

Noncompetitive inhibitors interact with free enzymes and ESCs with the same equilibrium constant [108]. For example, barbarin [11] and propanoic acid are both noncompetitive catecholase inhibitors of tyrosinase. 7,8,4′-trihydroxyflavone is a non-competitive tyrosinase inhibitor to diphenolase activity. Quenching analysis of the tyrosinase-7,8,4′-trihydroxyflavone complex has shown that tyrosinase has a single binding site for 7,8,4′-trihydroxyflavone via van der Waals forces and H-bonds. A molecular docking simulation of tyrosinase has shown that hydrogen bonds are crucial for the interaction of 7,8,4′-trihydroxyflavone with the active site His244/Met280. Polyphenols of flavonoids, phenolic acids, stilbenes, and lignans are tyrosinase inhibitors. Tricin is a noncompetitive tyrosinase inhibitor. CD spectra analysis has shown that tricin binding to tyrosinase causes a change in the secondary structure along with stereospecific changes in the conformation alterations of tyrosinase [109]. Tricin interacts with tyrosinase in amino acid residues in the hydrophobic pocket, as has been proven by fluorescence quenching. The tricin-enzyme complex is stabilized by hydrophobic interaction as well as hydrogen bonds with Asn80 and Arg267. The results of docking analysis have shown that the inhibition can be attributed to the stereospecific blockade of tricin in the tyrosinase active center. Interestingly, the 8-prenylkaempferol derivative Kushenol A is a noncompetitive tyrosinase inhibitor. Isoflavone glabridin is a noncompetitive inhibitor that has been shown to inhibit tyrosinase with a static mechanism [110], while a synthetic 3,4-dihydroglabridin has shown stronger inhibitory activity than glabridin. 4-Substituted resorcinol skeleton binding to the enzyme causes 3D conformational adaptation [111]. *p*-coumaric acid ethyl ester (*p*-CAEE) has been shown to inhibit tyrosinase in a noncompetitive manner. A docking simulation has implied that *p*-CAEE induces a conformational shift in the catalytic region to change the binding forces of L-tyrosinase [112]. Alkynyl glycoside analogues modified by the C-1 alkynyl and C-6 alkyl groups of the sugar ring have been shown to be a non-competitive mode inhibitor to the tyrosinase catalysis of L-tyrosine and L-DOPA that are comparable to arbutin and kojic acid. SARs analysis has determined that the C-6 alkyne and alkyl groups of the sugar ring and stereoisomer are the inhibitory points [113]. 

Caffeine is a noncompetitive tyrosinase inhibitor [114]. A molecular dynamics simulation has indicated that caffeine binds to the amino acids of the Gln307, Thr308, Asp312, Glu356, Asp357, Lys 376, Trp358, and Lys379 residues of the tyrosinase, thus causing tyrosinase inhibition via a conformational shift in the binding sites and the active center loop of L-tyrosinase. Verbascoside and 2,4-dihydroxy-1,4-benzoxazin-3-one are non-competitive inhibitors of tyrosinase [115]. Betulinic acid is also a non-competitive inhibitor, according to in silico analysis [116]. The tyrosinase has a specific binding region with polarity and non-polarity on the active site [116]. 3-*O*-α-l-rhamnosyl-2-gallate quercetin and biflavanols are tyrosinase inhibitors with specific inhibitory mechanisms. Among those compounds, potenserin-C/-D and 3-*O*-α-l-rhamnosyl-2-gallate quercetin have been shown to be reversible noncompetitive inhibitors by in silico computational molecular simulations and enzyme kinetics. Molecular simulation analysis has indicated that the galloyl substitution of the glucose moiety is essential for enzyme inhibition. Moreover, the use of 3,4,5-OH additives in the aglycones of flavonoids is required to inhibit tyrosinase activity. Biflavanols increase inhibitory activity via electrostatic interactions, hydrogen bonds, and π-alkyl bonds [117]. Polydatin is also a potent noncompetitive inhibitor of tyrosinase that is comparable to the control of kojic acid. Its tyrosinase inhibitory activity has been deduced in in silico molecular docking. The ionized (-)-8-chlorocatechin binds to tyrosinase with high and stable binding affinity to the catalytic site of the tyrosinase active site, while polydatin interacts with the enzymic allosteric sites that are predicted to be stronger than the catalytic site of tyrosinase [118].

From marine natural products such as Formosan soft coral *Cladiella australis*, 4-phenylsulfanyl butan-2-one inhibits tyrosinase activity in a non-competitive manner. It inhibits melanin production rather than the control compounds of 1-phenyl-2-thiourea and arbutin. 4-Phenylsulfanyl butan-2-one also inhibits melanin synthetic proteins such as glycoprotein 100, MITF, TRP-1, and DCT (or TRP-2) [119]. Similarly, 2-acetyl-5-methoxyphenyl-3-4-hydroxyphenyl acrylate acts as a non-competitive inhibitor [104]. Benzaldehyde is a partial noncompetitive inhibitor in the inhibition of 4-t-butylcatechol oxidation by tyrosinase. The C-4 positional substitution of 4-penthylbenzaldehyde shifts the benzaldehyde to a full- and mixed-type inhibitor of diphenolase activity. Therefore, benzaldehyde C-4-substitution of the aromatic backbone represents the hydrophobic interaction with the catalytic center of tyrosinase [120]. Oxyresveratrol has shown noncompetitive inhibition of diphenolase activity. Mulberroside A (MA) has shown competitive inhibition of diphenolase activity. However, MA and OR are respectively reversible- and mixed-type inhibitors of monophenolase activity [121]. In enzyme kinetic analysis, rottlerin has been shown to be a mixed-type inhibitor whereas mallotophilippen A and B have both been shown to be non-competitive-type inhibitors [122]. Diethyl 1-6-oxocyclohex-1-en-1-yl ethyl-phosphonate and 2-hydroxymethyl cyclohex-2-enone, as cyclic Morita-Baylis-Hillman adducts, are competitive inhibitors. However, 6-oxocyclohex-1-en-1-yl ethyl acetate is a non-competitive inhibitor. They strongly attenuate melanin biosynthesis, cellular enzyme activity, and protein expression of tyrosinase, thus resulting in strong inhibition of melanogenesis [123]. Table 4 summarizes the noncompetitive tyrosinase inhibitors.

## 4. Tyrosinase Inhibitor Types Based on Copper Chelation and Melanogenic Downregulation 

### 4.1. Properties of Metalloenzyme Tyrosinase

Metalloenzyme tyrosinase possesses two copper ions at the enzyme active site. Therefore, efficient chelators of metal ions can induce metal coordination for enzyme inhibition. Consequently, compounds with potential affinities for Cu(II), Fe(III), Al(III), and Zn(II) are industrially and pharmaceutically applicable for use in various settings. Tyrosinase protein binds to copper ions in an essential step for tyrosinase enzyme activity, folding correction, and protein maturation. Tyrosinase active sites possess binuclear copper ions as a determinant of the tyrosinase catalysis level; reversely, copper’s chelation by independent agents inhibits enzyme activity. A metal constituent’s copper ions are present at the active and catalytic domain of tyrosinase. Inhibitors bind to the dicopper(II) active site. Hydroxyl groups of the natural compounds can experimentally quench the intrinsic fluorescence in the tyrosinase enzyme. The natural products can directly interact with the Cu^2+^ ions, which can be proven by analytical approaches such as UV-visible spectroscopic observation for diverse red peak shifts, for example, at 278 nm to 295 nm red shift. The copper ion-bound inhibitors reflect chelate formation.

### 4.2. Copper Chelating Tyrosinase Inhibitors

Some compounds exhibit melanogenesis inhibition capacities by different inhibitory modes, including intracellular tyrosinase inhibitory activity, copper chelation, and melanin-associated protein downregulations such as MITF [124]. For example, casuarictin—as an ellagitannin compound—is a melanogenic inhibitor through the inhibition of intracellular tyrosinase activity and copper chelation as well as MITF protein downregulation [124]. 7,8,4-Trihydroxyflavone inhibits L-DOPA oxidation by tyrosinase in a reversible and non-competitive mode. It chelates copper ions to lead to tyrosinase complex formation within a single binding site via hydrogen bonding and van der Waals interactions [125]. Ellagic acid (EA) is a copper iron chelator and inhibits melanogenesis via autophagy induction, i.e., EA-induced autophagy. Similarly, vanillic acid (VA) is a tyrosinase inhibitor that inhibits the tyrosinase of monophenolase and diphenolase. VA is neither a copper chelator nor an inducer of changes in the enzyme conformation shift. VA binds to the dicopper-centered amino acid residues and hampers the interaction of tyrosinase with its substrates [126]. 

Among flavonols, galangin, kaempferol, and quercetin chelate copper in the enzyme and consequently inhibit tyrosinase activity, while related compounds such as apigenin, flavones, chrysin, and luteolin are not copper chelators of tyrosinase. However, the three flavonols of quercetin, galangin, and kaempferol commonly chelate copper ions and inhibit diphenolase activity in the enzyme but display different pathways. Thus, their inhibitory activities stem from copper chelation. The chelation mechanism of copper ions is also found in the known flavonols. Kaempferol is a common flavonol that is a competitive inhibitor of L-DOPA oxidation catalyzed by tyrosinase with an ID50 0.23 mM, while 3-*O*-glycosyl kaempferol is not a tyrosinase inhibitor. For example, the chelation mechanism is specific to all flavonols aside from their free 3-hydroxyl group. Flavonols have different inhibition mechanisms of tyrosinase activity. A flavonol called quercetin is a tyrosinase cofactor without monophenolase inhibition. Meanwhile, another flavonol, galangin, is a monophenolase inhibitor without cofactor action. Yet another flavonol called kaempferol is neither a tyrosinase cofactor nor a monophenolase inhibitor [127,128].

Condensed tannins are known to be reversible- and mixed-type inhibitors, but they have been reported to have a different inhibitory mode than the typical enzyme inhibition [129]. As condensed tannins chelate metal ions [130], when the condensed tannins reduce the *o*-quinones known as enzymic product, the tannins bind to the Cu^2+^ ions and the copper ion-bound inhibitors indicate the chelate formation. Tannin-copper chelate formation induces an interaction potential of the tannin with copper ions and the active site, thus causing tyrosinase inhibition. Fluorescence quenching analysis indicates that the condensed tannin hydroxyl groups quench the intrinsic fluorescence of tyrosinase. Pentagalloylglucose (PGG) is a potential tyrosinase inhibitor to monophenolase and diphenolase by blocking dopachrome formation and the strong chelation of copper ions. PGG interacts with the Glu-173, Glu-208, Lys-158, Lys-180, Gln-44, and Gln-159 residues of tyrosinase through hydrogen bonds [131]. Sesquiterpenoids of 1,8-cineole, α-terpineol, thymol, α-/β-phellandrene, (*E*)-nerolidol, and β-caryophyllene isolated from essential oils have been shown to inhibit tyrosinase in vitro when L-DOPA and L-tyrosine substrates were used in an in silico molecular docking study. They interact with copper ions with a bathochromic shift around 15 nm as well as between the oxygen and Cu^2+^ or His263/His259 residues of the enzyme active site near a location at less than four of the enzymes [132].

For peptides, diphenolase inhibitory peptides interact with aromatic amino acid residues and chelate copper ions within the active site adjacent to the substrate-binding pocket. The GYSLGNWVCAAK peptide acts as a competitive inhibitor with copper chelating activity [133]. Recently, from rice-bran albumin, the SSEYYGGEGSSSEQGYYGEG peptide has been shown to have effective tyrosinase inhibitory and copper-chelating activities. This exhibits dual potential as both a natural inhibitor of tyrosinase activity and a copper chelator [134]. From the fish scale polypeptides, a tyrosinase-inhibitory and copper ion-chelating peptide was isolated [135]. Tilapia skin-derived peptides PFRMY and RGFTGM inhibit tyrosinase enzyme and chelate copper ions through hydrogen bonds and hydrophobic interactions [136].

A thioamide/azepine, 5,6,7,8-tetrahydro-4H-furo 3,2-c-azepine-4-thione (T4FAT), is a copper chelator with melanogenic and tyrosinase activity inhibition. The action mechanism of T4FAT is of the non-competitive inhibitor type [137]. However, T4FAT attenuates tyrosinase protein expression but not mRNA transcription. Copper chelation and tyrosinase inhibition require the thioamide group present on T4FAT for subsequent melanogenesis inhibition. T4FAT and kojic acid, as copper chelators, are similar in their chelation properties, but T4FAT exhibits superior melanogenesis inhibition compared to kojic acid. Table 5 summarizes the chelating inhibitors of tyrosinase.

## 5. Conclusions

The main activator in melanin biosynthesis is the enzyme tyrosinase, so skin-whitening compounds are developed with the target of regulating tyrosinase enzyme activity or the melanogenesis-associated factor expressions in cells and tissues. Thus, scientists have synthesized and modified those natural forms to generate semi-synthetic and synthetic inhibitors. Most tyrosinase inhibitors consist of phenolics, polyphenolics, and their derivatives, as well as phenolic-backbone independent compounds, which are collectively referred to as phenolic compounds. They include azole, benzaldehyde, hydroxypyridinone, kojic acid, phenyl, piperidine, pyridine, pyridinone, terpenoid, thiosemicarbazide, thiosemicarbazone, thiazolidine, and xanthates. Among natural inhibitors, phenolic-backbone structures are the major inhibitors of tyrosinase. The present review has mainly focused on naturally occurring tyrosinase inhibitors and their derivatives. Among them, restricted sources are safe for human usage, as suggested by in vitro and in vivo models as well as molecular docking simulation experiments. Melanogenesis inhibition can be obtained by three different modes: tyrosinase inhibition, copper chelation, and melanin-associated protein downregulation. Tyrosinase inhibitors are of four types: competitive, uncompetitive, competitive/uncompetitive mixed, and noncompetitive inhibitors, which are differentiated solely based on their inhibition mechanisms. Some tyrosinase inhibitors show a reversible mode, thereby acting as a suicide substrate, as conceptional tyrosinase inhibitors are classified as inactivators and reversible solely based on the molecular recognizing properties of the enzyme. Tyrosinase inactivators or suicide inactivators induce 3D structural and conformational shifts toward enzyme inactivation. In a minor mechanism, transcription factors such as MITF can also be downregulated by inhibitors.

## Figures and Tables

**Figure 1 ijms-24-08226-f001:**
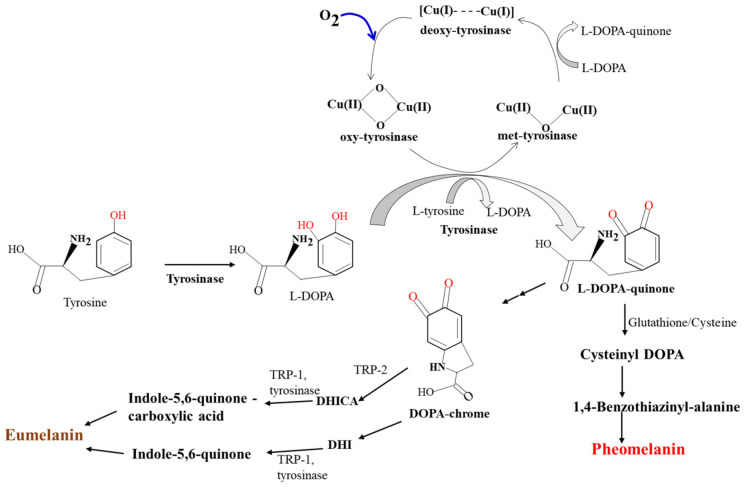
**Melanin biosynthesis. Tyrosinase uses L-tyrosine and L-DOPA as catalysis substrates**. *Deoxy*-tyrosinase binds O_2_ to yield *oxy*-tyrosinase to function as a monophenolase or diphenolase enzyme. The *met*-tyrosinase catalyzes diphenolase. Eumelanin ranges in color from black to brown. Pheomelanin ranges in color from yellow to reddish. The color of eumelanin is stronger than that of pheomelanin.

**Figure 3 ijms-24-08226-f003:**
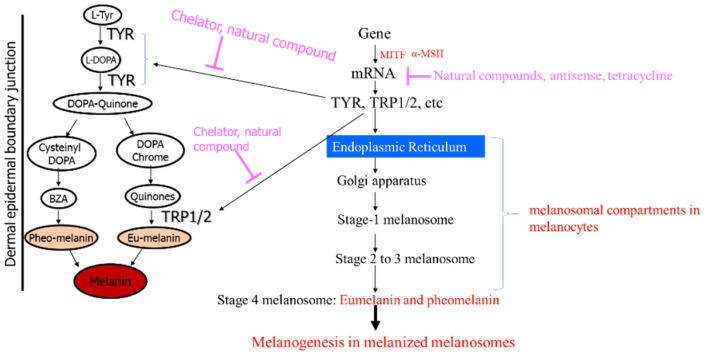
**Melanogenesis in melanized melanosomes and strategic approaches for reducing the level of melanin levels in dermal-epidermal junctional skin.** D-DOPA, L-3,4,-dihydroxyphenylalanine; BZA, 1,4-Benzothiazinyl-alanine; MITF, microphthalmia-associated transcription factor; α-MSH, α-melanocyte-stimulating hormone.

**Figure 4 ijms-24-08226-f004:**
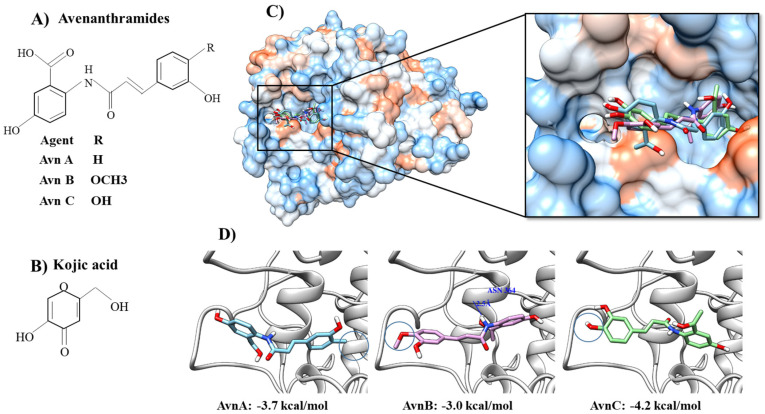
**Molecular docking simulation of avenanthramides with human tyrosinase**. (**A**) Avenanthramides. (**B**) Kojic acid. (**C**) Molecular docking simulation of avenanthramides (Avn A, B, and C). (**D**) Each calculated affinity expressed in kcal/mol. Detailed molecular docking of AvnA-C can be referenced in our previous publication (Park et al., 2021. [14]). Circles indicate the interaction atomic positions.

**Table 1 ijms-24-08226-t001:** Summary of the competitive tyrosinase inhibitors.

Compounds	Compound Class	Competitive Type	References
Hydroquinone	α-/β-arbutins		[1,26]
4-6-Hydroxy-2-naphthyl-1,3-bezendiol, resorcinol	resorcin		[13]
Anthocyanidin, aurone, flavan-3,4-diol	flavonoid		[28]
Kuwanon C, papyriflavonol A, sanggenon D, sophoflavescenol, flavonoid vinylation, lupinalbin, 7-*O*-gentibiosyl 2′-hydroxygenistein, gallocatechin, proanthocyanidins, (-)-8-Chlorocatechin, cyanidin, delphinidin, malvidin, pelargonidin, peonidin	flavonoid derivative		[29,36,47,52,53,56]
3-*O*-β-Galactosyl quercetin, 3′,5′-Di-*C*-β glucosyl phloretin, galactosyl-3-myricetin, potenserin C, 3-*O*-α-L-rhamnosyl quercetin-2-gallate	flavonoid glycoside		[30]
Apigenin, chrysin, luteolin, baicalein, mormin, cyclomorusin, morusin, norartocarpetin	flavone		[31,37]
Apigetrin, vitexin, baicalin, nobiletin, tangeretin, biflavone,7,8,4′-trihydroxy-isoflavone, 7,3′,4′-trihydroxy-isoflavone, 3-*O*-β-D-glucosyl 4,5,5,7,7-pentahydroxy 3,3-dimethoxy 3,4-*O*-biflavone, isovitexin, baicalein, 6-hydroxyapigenin, 6-hydroxy-kaempferol, 6-hydroxygalangin, tricin (5,7,4-trihydroxy-3,5-dimethoxyflavone), vitexin	flavone derivatives (C-glycosyl flavone, hydroxyflavone, flavone glucoside)	monophenolase/diphenolase	[32,33,34,35,36,40,42]
Quercetin, 4-*O*-β-D-glucosyl quercetin, β-*D*-glucosyl 3-*O*-6-*O*-malonyl quercetin, β-D-glucosyl 3-*O*-6-*O*-malonyl-kaempferol, morin, (±)2,3-cis-dihydromorin, 2,3-trans-dihydromorin, galangin, kaempferol, 8-prenylkaempferol, epicatechin, epigallocatechin, epicatechin gallate, epigallocatechin gallate, catechin, proanthocyanidin	flavonol	diphenolase	[38,39,50,51,54,55]
6,7,4′-Trihydroxyisoflavone	hydroxyflavone	monophenolase	[41]
Daidzein, glyasperin C, formononetin, genistein, mirkoin, texasin, tectorigenin, odoratin	Isoflavone		[45]
Eriodictyol, naringenin, hesperetin, hesperidin, liquiritin naringin, taxifolin, 6-isoprenoid flavanone, nigrasin K, sanggenon M/C/O, chalcomoracin, kuwanon J, sorocein H	flavanone		[48,49]
Curcumin, desmethoxycurcumin, hydroxybenzoate, hydoxycinnamate	phenolic compound		[57]
Esculetin, 8-epi-cleomiscosin, umbelliferone, thiophosphonic acid diamide, diazaphosphinane, resveratrol-hybrid	coumarin		[58,59,60,61]
Butein, chalcone, flavan-3,4-diols, dihydroflavone, dihydrochalcone, 1,3-diphenyl-2-propen-1-one, carthamin, phloretin, sappan-chalcone, isoliquiritigenin, glabrene, 2,4,2,4-hydroxycalcone, 2,4,2′,4′-tetrahydroxychalcone 2,4,2′,4′-tetrahydroxy-3-3-methyl-2-butenyl-chalcone, vulpinoideol-B, dihydrochalcone, morachalcone-A, bavachinin	chalcone		[34,62]
2-Arylbenzofuran, 2R-2,3-dihydro-2-1-hydroxy-1-methylethyl-2,6-bibenzofuran-6,4-diol, benzofuran flavonoid mulberrofuran G, albanol B, macrourins E	aurone		[65,66]
Resveratrol, oxyresveratrol, azo-resveratrol, azo-oxyresveratrol, E-2-2,4-dihydroxyphenyl, diazinyl, phenyl-4-methylbenzenesulfonate, trans-resveratrol, resveratrol dimer gnetin-C, hydroxystillbene	stilbenes		[56,73,74,75,76,77]
Monoterpenoid phenol, carvacrol aand its derivatives, bakuchiol, iridoid glucoside, sylvestrosyl 7-*O*-caffeoyl-I, sylvestrosyl 7-*O*-*p*-coumaroyl-I	terpenoid		[79]

**Table 2 ijms-24-08226-t002:** Summarization of the uncompetitive tyrosinase inhibitors.

Compounds	Compound Class	Uncompetitive Type	References
Deoxyarbutin			[1]
Luteolin		diphenolase	[87]
2,5-Dihydroxyacetophenone (DHAP), 2,6-DHAP			[90]
β-D-Glucosyl 3,4-dihydroxy-5-methoxybiphenyl-2-*O*			[91]
β-D-Glucosyl Ov-16-4-3,4-dihydroxybenzoyloxymethyl phenyl-*O*	phenolic glycoside		[92]
Sophorcarpidine	flavonoid glycoside		[30]

**Table 3 ijms-24-08226-t003:** Summarization of the mixed competitive and uncompetitive tyrosinase inhibitors.

Compounds	Compound Class	Mixed Type	References
Cinnamic acid, aloin, hydroxypyridinone derivatives, phthalic acid derivatives			[93]
D-Arabinose, brazilein and thymol derivatives		diphenolase	[96,97]
Baicalein			[98]
Proanthocyanidin, procyanidin, prodelphinidin, propelargonidin, and the acyl derivatives (galloyl benzoate, *p*-hydroxybenzoate	tannin		[99]
3-Phenylbenzoic acid	phenolic acid		[100]
2-*S*-Lipoyl-CA	CA-dihydrolipoic acid *S*-conjugate		[101]
Isosilybin A/B, silydianin, 2,3-dihydrosilychristin, silybin, silychristin-A/-B	flavonolignan	monophenolase/diphenolase	[102]

**Table 4 ijms-24-08226-t004:** Summarization of the noncompetitive tyrosinase inhibitors.

Compounds	Compound Class	Noncompetitive Type	References
Barbarin, propanoic acid			[11]
7,8,4-Trihydroxyflavone		diphenolase	[109]
8-Prenylkaempferol derivative Kushenol A, glabridin, 3,4-dihydroglabridin	isoflavone		[110]
p-Coumaric acid ethyl ester			[112]
4-Substituted resorcinol			[111]
Alkynyl glycoside analogues			[113]
Caffeine			[114]
Verbascoside and 2,4-dihydroxy-1,4-benzoxazin-3-one			[115]
Betulinic acid			[116]
3-*O*-α-l-Rhamnosyl-2-gallate quercetin, biflavanols, potenserin-C/-D, 3-*O*-α-l-rhamnosyl-2-gallate quercetin, biflavanol		reversible/noncompetitive	[116,117]
Polydatin, (-)-8-chlorocatechin, polydatin			[118]
4-Phenylsulfanyl butan-2-one, 2-acetyl-5-methoxyphenyl-3-4-hydroxyphenyl acrylate, benzaldehyde	marine natural products	diphenolase	[104,119,120]
Oxyresveratrol, mulberroside A	marine natural products	diphenolase	[121]
Mallotophilippen A, B	marine natural products	monophenolase	[122]
6-Oxocyclohex-1-en-1-yl ethyl acetate	marine natural products		[123]

**Table 5 ijms-24-08226-t005:** Summarization of the chelating tyrosinase inhibitors.

Compounds	Compound Class	Type	References
Casuarictin	Ellagitannin	noncompetitive	[124]
7,8,4-Trihydroxyflavone		reversible/non-competitive	[125]
Ellagic acid		monophenolase/diphenolase	[126]
Vanillic acid		monophenolase/diphenolase	[126]
Galangin, kaempferol, quercetin	flavonol	competitive, monophenolase	[127,128]
Pentagalloylglucose	tannin	monophenolase/diphenolase	[129,130,131]
1,8-Cineole, α-terpineol, thymol, α-/β-phellandrene, (E)-nerolidol, β-caryophyllene	sesquiterpenoid/essential oil		[132]
GYSLGNWVCAAK	peptide	competitive	[133]
SSEYYGGEGSSSEQGYYGEG	peptide		[134]
PFRMY, RGFTGM	peptide	reversible/noncompetitive	[136]
5,6,7,8-Tetrahydro-4H-furo 3,2-c-azepine-4-thione	thioamide/azepine		[137]

## Data Availability

Data sharing is available on request; schematic data is not applicable to this article as no new data were created or analyzed in this study.

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
