# Peer review of "Naturally-Occurring Tyrosinase Inhibitors Classified by Enzyme Kinetics and Copper Chelation"

_ijms, 2023, doi:10.3390/ijms24098226_

Round 1

Reviewer 1 Report

Dear Authors, please carefully check the content for writing style, chemical names, chemical structures, abbreviations, and English. Few examples given below:

1- The chemical structures and chemical names should be revised and corrected.

2- A table with the activity value(s) for each of every compound listed needs to be constructed.

3- Discussion of the activity within the standard activity margins needs to be discussed.

4- Classification of the inhibitors according to the chemical class and SARS will be helpful/interesting to the reader.

5- The authors need to be careful and match between different names of the same molecule (lines 129/135; vitamin C = L-ascorbic acid), and correctly discuss the same, also in line 134, arbutin was mentioned twice.

6- The paragraph, between lines 132-137, needs careful re-phrasing.

7- Line 150, please correct and re-phrase.

8- Line 154, check the typos.

9- please write L-DOPA correctly.

10- line 156, please correct the chemical name, and many more in the text.

Author Response

A) Review 1

1- The chemical structures and chemical names should be revised and corrected.

   As suggested, the most compound structures are known. Chemical names are corrected with the additional Table 1-5.

2- A table with the activity value(s) for each of every compound listed needs to be constructed.

As suggested, the new Tables 1 to -5 have been constructed.

In line 370,

Table 1. Summarization of the competitive tyrosinase inhibitors

Compounds

Property

Competitive type

Reference

Hydroquinone

α-/β-arbutins

1,26

4-6-Hydroxy-2-naphthyl-1,3-bezendiol, resorcinol

resorcin

13

Anthocyanidin, aurone, flavan-3,4-diol

flavonoid

28

Kuwanon C, papyriflavonol A, sanggenon D, sophoflavescenol, flavonoid vinylation, lupinalbin, 7-O-gentibiosyl 2′-hydroxygenistein, gallocatechin, proanthocyanidins, (-)-8-Chlorocatechin, cyanidin, delphinidin, malvidin, pelargonidin, peonidin

flavonoid derivative

29,47,36,

52,53,56

3-O-β-Galactosyl quercetin, 3',5'-Di-C-β glucosyl phloretin, galactosyl-3-myricetin, potenserin C, 3-O-α-L-rhamnosyl quercetin-2-gallate

flavonoid glycoside

30

Apigenin, chrysin, luteolin, baicalein, mormin, cyclomorusin, morusin, norartocarpetin

flavone

31,37

Apigetrin, vitexin, baicalin, nobiletin, tangeretin, biflavone,7,8,4'-trihydroxy-isoflavone, 7,3',4'-trihydroxy-isoflavone, 3-O-β-D-glucosyl 4,5,5,7,7-pentahydroxy 3,3-dimethoxy 3,4-O-biflavone, isovitexin, baicalein, 6-hydroxyapigenin, 6-hydroxy-kaempferol, 6-hydroxygalangin, tricin (5,7,4-trihydroxy-3,5-dimethoxyflavone), vitexin

flavone derivatives (C-glycosyl flavone, hydroxyflavone, flavone glucoside)

monophenolase/

diphenolase

32,33,34,

35,36,40,42

Quercetin, 4-O-b-D-glucosyl quercetin, β-D-glucosyl 3-O-6-O-malonyl quercetin, β-D-glucosyl 3-O-6-O-malonyl-kaempferol, morin, (±)2,3-cis-dihydromorin, 2,3-trans-dihydromorin, galangin, kaempferol, 8-prenylkaempferol, epicatechin, epigallocatechin, epicatechin gallate, epigallocatechin gallate, catechin, proanthocyanidin

flavonol

diphenolase

38,39,50,

51,54,55

6,7,4'-Trihydroxyisoflavone

hydroxyflavone

monophenolase

41

Daidzein, glyasperin C, formononetin, genistein, mirkoin, texasin, tectorigenin, odoratin

Isoflavone

45

Eriodictyol, naringenin, hesperetin, hesperidin, liquiritin naringin, taxifolin, 6-isoprenoid flavanone, nigrasin K, sanggenon M/C/O, chalcomoracin, kuwanon J, sorocein H

flavanone

48,49

Curcumin, desmethoxycurcumin, hydroxybenzoate, hydoxycinnamate

phenolic compound

57

Esculetin, 8-epi-cleomiscosin, umbelliferone, thiophosphonic acid diamide, diazaphosphinane, resveratrol-hybrid

coumarin

58-61

Butein, chalcone, flavan-3,4-diols, dihydroflavone, dihydrochalcone, 1,3-diphenyl-2-propen-1-one, carthamin, phloretin, sappan-chalcone, isoliquiritigenin, glabrene, 2,4,2,4-hydroxycalcone, 2,4,2′,4′-tetrahydroxychalcone 2,4,2',4'-tetrahydroxy-3-3-methyl-2-butenyl-chalcone, vulpinoideol-B, dihydrochalcone, morachalcone-A, bavachinin

chalcone

34,62

2-Arylbenzofuran, 2R-2,3-dihydro-2-1-hydroxy-1-methylethyl-2,6-bibenzofuran-6,4-diol, benzofuran flavonoid mulberrofuran G, albanol B, macrourins E

aurone

65,66

Resveratrol, oxyresveratrol, azo-resveratrol, azo-oxyresveratrol, E-2-2,4-dihydroxyphenyl, diazinyl, phenyl-4-methylbenzenesulfonate, trans-resveratrol, resveratrol dimer gnetin-C, hydroxystillbene

stilbenes

56,73-77

Monoterpenoid phenol, carvacrol aand its derivatives, bakuchiol, iridoid glucoside, sylvestrosyl 7-O-caffeoyl-I, sylvestrosyl 7-O-p-coumaroyl-I

terpenoid

79

In line 397,

Table 2. Summarization of the uncompetitive tyrosinase inhibitors

ompounds

Property

Uncompetitive type

Reference

Deoxyarbutin

1,

Luteolin

diphenolase

87

2,5-Dihydroxyacetophenone (DHAP), 2,6-DHAP

90

β-D-Glucosyl 3,4-dihydroxy-5-methoxybiphenyl-2-O

91

β-D-Glucosyl Ov-16-4-3,4-dihydroxybenzoyloxymethyl phenyl-O

phenolic glycoside

92

Sophorcarpidine

flavonoid glycosides

30

In line 427,

Table 3. Summarization of the mixed competitive and uncompetitive tyrosinase inhibitors

Compounds

Property

Mixed type

Reference

Cinnamic acid, aloin, hydroxypyridinone derivatives, phthalic acid derivatives

93

D-Arabinose, brazilein and thymol derivatives

diphenolase

96,97

Baicalein

98

Proanthocyanidin, procyanidin, prodelphinidin, propelargonidin, and the acyl derivatives (galloyl benzoate, p-hydroxybenzoate

tannin

99

3-Phenylbenzoic acid

pheloic acid

100

2-S-Lipoyl-CA

CA-dihydrolipoic acid S-conjugate

101

Isosilybin A/B, silydianin, 2,3-dihydrosilychristin, silybin, silychristin-A/-B

flavonolignan

monophenolase/diphenolase

102

In line 483,

Table 4. Summarization of the noncompetitive tyrosinase inhibitors

Compounds

Property

Noncompetitive type

Reference

Barbarin, propanoic acid

noncompetitive

11

7,8,4-Trihydroxyflavone

diphenolase

109

8-Prenylkaempferol derivative Kushenol A, glabridin, 3,4-dihydroglabridin

isoflavone

110

p-Coumaric acid ethyl ester

112

4-Substituted resorcinol

111

Alkynyl glycoside analogues

113

Caffeine

114

Verbascoside and 2,4-dihydroxy-1,4-benzoxazin-3-one

115

Betulinic acid

116

3-O-α-l-Rhamnosyl-2-gallate quercetin, biflavanols, potenserin-C/-D, 3-O-α-l-rhamnosyl-2-gallate quercetin, biflavanol

reversible/noncompetitive

116,117

Polydatin, (-)-8-chlorocatechin, polydatin

1118

4-Phenylsulfanyl butan-2-one, 2-acetyl-5-methoxyphenyl-3-4-hydroxyphenyl acrylate, benzaldehyde

marine natural products

diphenolase

104,119,120

Oxyresveratrol, mulberroside A

marine natural products

diphenolase

121

Mallotophilippen A, B

marine natural products

monophenolase

122

6-Oxocyclohex-1-en-1-yl ethyl acetate

marine natural products

123

In line 483,

Table 4. Summarization of the noncompetitive tyrosinase inhibitors

Compounds

Property

Noncompetitive type

Reference

Barbarin, propanoic acid

noncompetitive

11

7,8,4-Trihydroxyflavone

diphenolase

109

8-Prenylkaempferol derivative Kushenol A, glabridin, 3,4-dihydroglabridin

isoflavone

110

p-Coumaric acid ethyl ester

112

4-Substituted resorcinol

111

Alkynyl glycoside analogues

113

Caffeine

114

Verbascoside and 2,4-dihydroxy-1,4-benzoxazin-3-one

115

Betulinic acid

116

3-O-α-l-Rhamnosyl-2-gallate quercetin, biflavanols, potenserin-C/-D, 3-O-α-l-rhamnosyl-2-gallate quercetin, biflavanol

reversible/noncompetitive

116,117

Polydatin, (-)-8-chlorocatechin, polydatin

1118

4-Phenylsulfanyl butan-2-one, 2-acetyl-5-methoxyphenyl-3-4-hydroxyphenyl acrylate, benzaldehyde

marine natural products

diphenolase

104,119,120

Oxyresveratrol, mulberroside A

marine natural products

diphenolase

121

Mallotophilippen A, B

marine natural products

monophenolase

122

6-Oxocyclohex-1-en-1-yl ethyl acetate

marine natural products

123

were newly written.

3- Discussion of the activity within the standard activity margins needs to be discussed.

As described, the standard activity has partly been discussed because the tyrosinase activity inhibitors are well reviewed in many different publications.

4- Classification of the inhibitors according to the chemical class and SARS will be helpful/interesting to the reader.

As described, Classification of the inhibitors according to the chemical class and SARS have well been reviewed in many independent review articles. However, the present review aims to summarize its inhibitory mechanism(s)-based classification.

5- The authors need to be careful and match between different names of the same molecule (lines 129/135; vitamin C = L-ascorbic acid), and correctly discuss the same, also in line 134, arbutin was mentioned twice.

As described, vitamin C has been corrected to L-ascorbic acid through the text. hydroquinone arbutin has been deleted.

6- The paragraph, between lines 132-137, needs careful re-phrasing.

As described, the lines 132-137 has been re-phrased.

Tyrosinase inhibitors have been purified from plants and kinetic tyrosinase inhibition studies using mushroom tyrosinases.

    Currently known skin-whitening and anti-melanin molecules include arbutin, deoxyarbutin, hydroquinone, deoxyarbutin derivatives, resorcinol, vanillin, niacinamide, kojic acid, arbutin-mimic isotachioside, hydroquinon deribvatives (α and β-arbutin), azelaic acid, L-ascorbic acid, ellagic acid and tranexamic acid

7- Line 150, please correct and re-phrase.

As described, the line 150 has been corrected.

 Inhibition of melanin biosynthesis reflects therapeutic and preventive options for melanogenesis. Tyrosinase inhibitors from natural products bind to the tyrosinases and the monophenolic tyrosinase (Fig. 3). Strategic approaches for reducing the level of melanin levels in dermal-epidermal junctional skin have been described with different ways of direct enzyme inhibitors and gene downregulators.

8- Line 154, check the typos.

As described, the line 154 has been corrected. Tyrosinase uses L-tyrosine and diphenolic L-DOPA as substrates

9- please write L-DOPA correctly.

As described, l-DOPA has been corrected to diphenolic L-DOPA

10- line 156, please correct the chemical name, and many more in the text.

As described, lone 56 has been corrected.

The met-tyrosinase catalyzes as diphenolase that oxidizes diphenol like L-DOPA to quinone and monophenolase that oxidizes monophenol like L-tyrosine to quinone like L-dopaquinone, although tyrosinase activity is generally measured using its substrate L-DOPA.

Author Response

Thank you for the reviewer 2 for his(her) careful reading of out manuscript. The present review mainly focuses on the action mechanisms of tyrosinase inhibitors, but not introductory review because there are numerous reviews available from the literature.

Line 19. In vitro has been italicized to “

Line 60. “Procucr” has been corrected to “produce”

Figure 1: it has been re-figured to a new one

Line 98. “Whitening agent development is consists of” has been corrected to “Whitening agent development consists of”

Line 100. “Cu2+” has been corrected to “Cu2+

Line 101. “NO” and “GM-GSF” in line 102

Line 122. via has been italicized to

Line 154. I-DOPA” has been corrected to “L-DOPA”

The writing has been coherent through the manuscript.

Line 228. “inhibitorory” has been corrected to “inhibitory”

Line 351. The bold “terpenoid” has been corrected to “terpenoid”.

Main molecular structures have been drawn

Line 515. “in silico” has been italicized to

Line 540. “in vitro” has been italicized to

Round 2

Reviewer 1 Report

Dear Authors, thanks for the correction, please check the comments below:

Figure 1. contains the wrong structures and should be re-drawn correctly

The chemical names should be corrected, C, p and O should be in italics (such as 3-O-β- 251 galactosyl quercetin, 3',5'-Di-C-β glucosyl phloretin // sylvestrosyl 7-O-p-coumaroyl-I), please check Table 1 as well.

table 1, the second column should be the class of compounds, not property. Also, it will be good to classify the structures according to the function of the class of compound. the current presentation is random. please check the typos in the table.

Tables 2, 3 and 4 contain typos and wrong and incomplete names

Author Response

2nd Responses to Reviewer and corrections

Dear Reviewers and Editor

(Separately uploaded 2nd response is available from the files)

Thank you for your careful checking of our revised version. We have carefully checked the content for writing style, chemical names, chemical structures, abbreviations, and English.

Questions

Figure 1. contains the wrong structures and should be re-drawn correctly

My Answer:

I am sorry for the tyrosine-modified structures and slightly corrected with redrawing..

The chemical names should be corrected, C, p and O should be in italics (such as 3-O-β- 251 galactosyl quercetin,

My Answer:

I am sorry for the uncareful writing in the compounds. The compound names are re-checked. For example, methoxy-hydroquinone-1-O-bD-glucopyranoside, 3-O-β-galactosyl quercetin, 3',5'-Di-C-β glucosyl phloretin and galactosyl-3-myricetin, Potenserin C and 3-O-α-L-rhamnosyl quercetin-2-gallate, 4'-O-b-D-glucosyl quercetin, β-D-glucosyl 3-O-6-O-malonyl quercetin, β-D-glucosyl 3-O-6-O-malonyl-kaempferol, sylvestrosyl 7-O-caffeoyl-I, sylvestrosyl 7-O-p-coumaroyl-I, , p-hydroxybenzoate, 3-O-α-l-Rhamnosyl-2-gallate quercetin, biflavanols, potenserin-C/-D, 3-O-α-l-rhamnosyl-2-gallate quercetin

table 1, the second column should be the class of compounds, not property. Also, it will be good to classify the structures according to the function of the class of compound. the current presentation is random. please check the typos in the table.

My Answer: As suggested, the property has been corrected to the Compound class through the Table 1- 5.

Tables 2, 3 and 4 contain typos and wrong and incomplete names

My Answer: As suggested, the typos and names have been re-checked. The examples: flavonoid glycoside (Table 2). phenolic acid (Table 3), noncompetitive removed (Table 4).

Editorial Question of Graphic abstract correction.

  As suggested, the Graphic abstract has been redrawed.
